# Phase transitions in few-monolayer spin ice films

L. Bovo [1,2], C.M. Rouleau[3], D. Prabhakaran[4] & S.T. Bramwell[1]

Vertex models are an important class of statistical mechanical system that admit exact solutions and exotic physics. Applications include water ice, ferro- and antiferro-electrics, spin ice and artificial spin ice. Here we show that it is possible to engineer spin ice films with atomic-layer precision down to the monolayer limit. Specific heat measurements show that these films, which have a fundamentally different symmetry to bulk spin ice, realise systems close to the two-dimensional F-model, with exotic phase transitions on topologically-constrained configurational manifolds. Our results show how spin ice thin films can release the celebrated Pauling entropy of spin ice without an anomaly in the specific heat. They also significantly expand the class of vertex models available to experiment.

[1] London Centre for Nanotechnology and Department of Physics and Astronomy, University College London, 17-19 Gordon Street, London WC1H 0AJ, UK. [2] Department of Innovation and Enterprise, University College London, 90 Tottenham Court Road, London W1T 4TJ, UK. [3] Oak Ridge National Laboratory, Center for Nanophase Materials Sciences, Chestnut Ridge Drive, Oak Ridge, TN 37831, USA. [4] Clarendon Laboratory, Department of Physics, University of Oxford, Parks Road, Oxford OX1 3PU, UK. Correspondence and requests for materials should be addressed to L.B. (email: l.bovo@ucl.ac.uk) or to S.T.B. (email: s.t.bramwell@ucl.ac.uk)

Vertex models were first introduced as statistical mechanical models of water ice and hydrogen bonded ferro- and antiferro-electrics[1,2], but they rapidly assumed a much broader importance as model many-body systems that admit exact analysis[3–7]. Their theoretical study[8] has led to many insights, including phase transitions on constrained manifolds[2,3], continuously variable critical exponents[7] and quantum to classical mappings[9]. Experimental realisations of these theoretical models were slower to follow, but the discovery[10–13] of spin ice and the development of artificial spin ice micromagnetic arrays[14–16] have significantly enlarged the set of real systems that closely approximate vertex models[17,18]. Such systems also have a general relevance to the physics of highly correlated spin liquid[19] and quantum spin ice[20] states, as they provide experimental examples of fractionalised excitations[21] and Coulomb phases[22].

The defining property of a vertex model is a set of vertex weights, which in experiment is determined by symmetry and tuned by temperature, pressure or applied fields. Spin ice, which consists of a cubic array of corner-linked spin tetrahedra, maps to a 16-vertex model, with the vertices represented by spin configurations on a tetrahedron[11]. Low temperature (in zero applied field) selects six degenerate vertices, making spin ice a realisation of Pauling's model of water ice[1], with extensive residual entropy[12]. In fact, the physics of bulk spin ice is even richer than that of a conventional 16-vertex model: the long-range part of the dipole–dipole interaction, which is significant in spin ice, is largely self-screened in the six-vertex spin ice state[23,24], but manifests as a magnetic Coulomb interaction between excited vertices to give the fractionalised magnetic monopoles of spin ice[21,25]. The discovery[21,23–25] that the addition of the long-ranged and conditionally convergent dipole–dipole interaction to a conventional vertex model resulted in such sharply-defined, exotic and realisable physics was a surprise that has stimulated considerable interest and activity in the field[22,26–33].

Recently, the first thin films of spin ice have been prepared[34,35]. We introduced ultra-thin epitaxial films of the spin ice material $Dy_2Ti_2O_7$ (DTO) grown on the isomorphic pyrochlore substrate $Y_2Ti_2O_7$ (YTO)[34]. This choice of substrate led to a very high degree of epitaxy with compressive strain perpendicular to the growth direction. The strain was found to remain homogenous even at relatively high thicknesses—certainly up to 60 nm[34]. Spin ice properties were observed at temperatures $T > 2$ K. However, at lower temperatures (down to 0.35 K) specific heat and magnetisation measurements showed that the Pauling entropy was completely released, although no anomaly was observed in the specific heat[34].

In the present work, we have further improved the fabrication of the thin films, such that the films are grown with atomic-layer precision along different crystallographic directions and have negligible interface effects. Owing to the homogenous strain, these films may realistically be regarded as low-dimensional variants of spin ice that have a reduced symmetry compared to that of the bulk. This raises the prospect of engineering and experimentally studying more general microscopic vertex systems than those afforded by bulk spin ice.

## Results

**Growth and characterisation of spin ice thin films.** Different schemes for altering vertex weights may be realised by growing spin ice films along different crystallographic directions: here we choose the cubic [100], [110] and [111] directions, respectively. Table 1 reports the samples that were prepared, separated into the three sets DTO||YTO(100), DTO||YTO(110) and DTO||YTO (111); within each set several different layer thicknesses were prepared. Note that, in the bulk cubic structure of pyrochlores (space group $Fd\bar{3}m$, lattice constant $a = 1.013$ nm for DTO, 1.0083 nm for YTO), 1 nm corresponds to ca. 1 monolayer (ML).

A layer-by-layer growth mechanism was confirmed by following the presence of a characteristic intensity oscillation in RHEED (reflection high-energy electron diffraction). This mechanism is similar to that we recently reported for the pyrochlore magnet $Tb_2Ti_2O_7$[36]. Figure 1a is an example of three RHEED patterns that were recorded during the deposition of thin films with different out-of-plane crystallographic directions. In the case of DTO||YTO(110) and DTO||YTO(111), the oscillation period measured at the specular reflection turns out to be compatible with the formation of a quarter of the unit cell along the out-of-plane direction, [110] and [111], respectively: here one RHEED oscillation corresponds to the formation of one tetrahedral layer of the pyrochlore structure. The thickness calculated from this model is in excellent agreement with the X-ray powder diffraction (XRD) analysis described below and reported in Table 1. In contrast, samples of DTO||YTO(100) present a different RHEED pattern with only one big oscillation of period roughly four times bigger than the other two sets; this is compatible with the creation of a complete unit cell. This tendency to grow in layers of unit cells rather than tetrahedra

**Table 1 $x$DTO||YTO($hkl$) list of samples, with $x$ indicating the number of monolayers and ($hkl$) the out-of-plane growth orientation**

| Sample | Pulses | Thickness (fringes) (nm) | Thickness (reflectivity) (nm) | Thickness (RHEED) (nm) | Density (reflectivity) (g cm$^{-3}$) |
|---|---|---|---|---|---|
| YTO | Annealed substrate | - | - | - | - |
| $x$DTO||YTO(100) | | | | | |
| $x = 44$ | 7332 | 43.5 (1) | 43.5 (1) | N/A | 7.1 (1) |
| $x = 8$ | 1222 | 7.8(1) | 7.9 (1) | N/A | 7.1 (1) |
| $x = 4$ | 611 | 3.9 (1)[a] | 4.0 (1) | N/A | 7.1 (1) |
| $x$DTO||YTO(110) | | | | | |
| $x = 44$ | 7332 | 44.8 (1) | 45.2 (1) | 44.5 (1) | 7.0 (1) |
| $x = 8$ | 1222 | 7.6(1) | 7.6 (1) | 7.6 (1) | 7.0 (1) |
| $x = 4$ | 611 | 3.9 (1)[a] | 4.2 (1) | 3.8 (1) | 7.0 (1) |
| $x$DTO||YTO(111) | | | | | |
| $x = 60$ | 9564 | 57.0 (1) | 57.1 (1) | 57.3 (1) | 7.2 (1) |
| $x = 10$ | 1594 | 9.5 (1) | 9.4 (1) | 9.6 (1) | 7.5 (1) |
| $x = 5$ | 797 | 4.8 (1)[a] | 4.8 (1) | 4.8 (1) | 7.4 (1) |

N/A not available, DTO $Dy_2Ti_2O_7$, YTO $Y_2Ti_2O_7$
[a]This thickness value has been extrapolated from the number of pulses

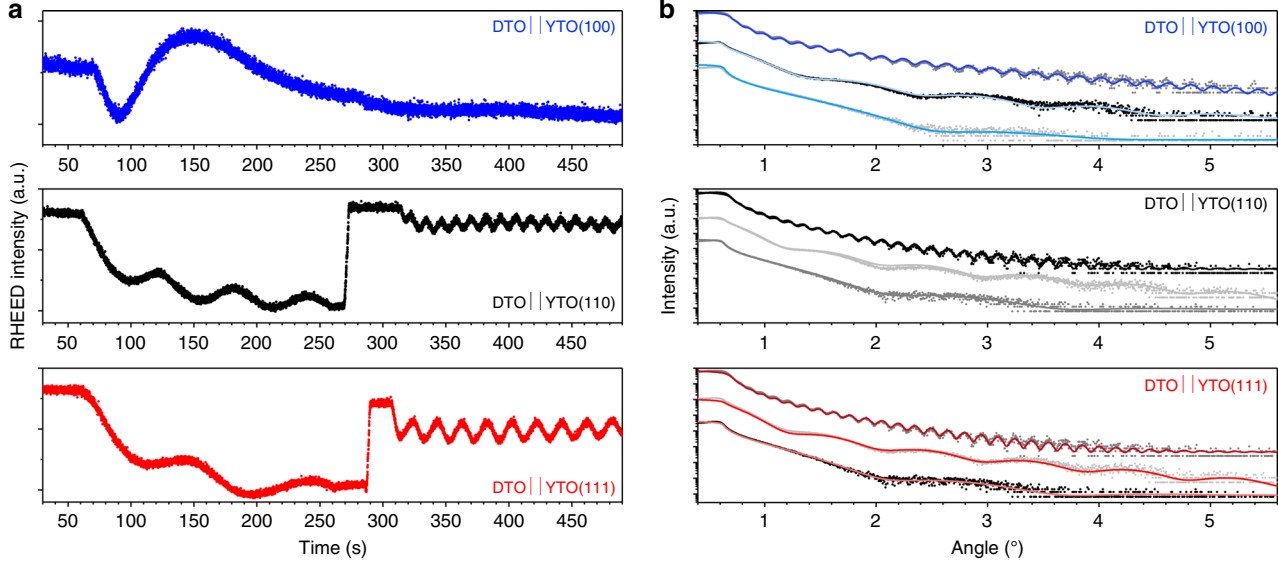

**Fig. 1** Layer-by-Layer growth and thickness control. Colour code: DTO||YTO(*hkl*) with (*hkl*) = (100) (blue), (*hkl*) = (110) (black), (*hkl*) = (111) (red). **a** Reflection high-energy electron diffraction (RHEED) oscillations measured at the specular spot during deposition. Three different growths are shown, along different crystallographic orientation. Two repetition rates were used during each deposition (initial 200 pulses at 1 Hz to help nucleation, increased to 5 Hz afterwards), hence the two oscillation periods. For samples DTO||YTO(110) and DTO||YTO(111), one oscillation corresponds to ¼ of the cell, equivalent to the formation of one tetrahedron of Dy ions (one unit cell along these orientations contains four corner shared tetrahedra of Dy ions). In the case of DTO||YTO(100) only one oscillation is detected, compatible with the formation of 1 unit cell (along this orientation growth, there is no stacking of corner shared tetrahedra of Dy ions). **b** Reflectivity measurements for all the *x*DTO||YTO(*hkl*) samples and related fits (values are reported in Table 1). In these reflectivity plots, each scan (samples with same *hkl* orientation, but different number of monolayers *x*) has been displaced vertically to improve clarity. DTO Dy$_2$Ti$_2$O$_7$, YTO Y$_2$Ti$_2$O$_7$

suggests that ⟨100⟩ faces grow relatively quickly, as implied by the fact (see ref. [13], Fig. 2) that flux-grown crystals show ⟨111⟩ faces (i.e. when grown in the absence of significant external forces crystals tend to eliminate their fastest growing faces). It also seems consistent with the observation that crystals grown by the floating-zone technique (which encourages a particular growth direction) tend to show ⟨110⟩ preferential growth axes[37,38].

Figure 1b displays X-ray reflectivity curves measured for each sample, as well as their relative fitting, the results of which are summarised in Table 1. The reflectivity analysis was used to estimate the thickness and the density of the film, both of which are expected to be reliable numbers in our analysis. The estimated total thicknesses are in close agreement with the values calculated on the basis of the number of laser pulses used during the growth. Furthermore, the density of the film is close to the tabulated bulk density of DTO (Table 1).

Epitaxy and compressive strain in the films were assessed by measuring the out-of-plane reflections (400), (440) and (444), as shown in Fig. 2. The out-of-plane lattice parameters are all compatible with 1.018(1) and 1.009(1) nm for the films and substrate, respectively. The thickness of each sample was determined from the fringes observed in the XRD pattern, with related values reported in Table 1. Two orthogonal off-specular reflections were measured as illustrated in Fig. 2 (panels b and c) for the thickest sample of each series. These measurements show that the films are very uniform (clear fringes appear in both spectra) and that they have a compressive strain, consistent with the smaller lattice constant of the substrate. In all three film orientations, analysis of the diffraction data indicates that the epitaxial strain is homogenous, the film structure being compressed equally within the in-plane directions, with the average density maintained by elongation along the out-of-plane direction. All our studies have indicated that the distortion does not depend on film thickness.

The epitaxial strain-induced lattice distortion affects the distance between the Dy ions and their oxide environment (hence the magnetic moments, and spin hamiltonian). The distortion is different depending on the chosen out-of-plane orientations. Figure 2d shows a schematic of one Dy tetrahedron and its orientation with respect to the cubic pyrochlore axis. In the cubic phase, all Dy–Dy distances are equal: $d_{Tb}$ = 3.5815 Å for a lattice constant of 1.013 nm. In the idealised film structure, not all the Dy–Dy distances are equivalent, as detailed in Table 2 and depicted in Fig. 2d. The angles between the tetrahedron edges are perturbed accordingly.

With homogenous strain, the space group symmetry is reduced to orthorhombic, A1, with a six-fold increase in the number of atoms per unit cell. A complete analysis of the crystal structure would be a challenging project for synchrotron X-ray techniques. However, despite the large unit cell, it is reasonable to expect a single local environment for the Dy$^{3+}$ ions, and our analysis of the magnetic and thermal properties below makes this approximation.

A commercial Physical Properties Measurement System (PPMS) (see Methods) was used to measure the heat capacity of the DTO thin films. This takes advantage of our previous demonstration[34,36,39] that the PPMS can be used to accurately measure the heat capacity of the thin films. To estimate the magnetic specific heat of the film, $C_M$, the measured heat capacity was corrected for the contributions from substrate and phonons using the same method described at length in ref. [34].

The thickest sample of each series, with 44 or 60 ML, that is, 44DTO||YTO(100), 44DTO||YTO(110) and 60DTO||YTO(111), were chosen as a prototype to present the specific heat behaviour of the epitaxial layers. The thinner films are discussed subsequently and also reported in Supplementary Figure 1 and Supplementary Figure 2. The heat capacity was found to scale closely with thickness of the films, as previously reported[34]. However, it must be noted

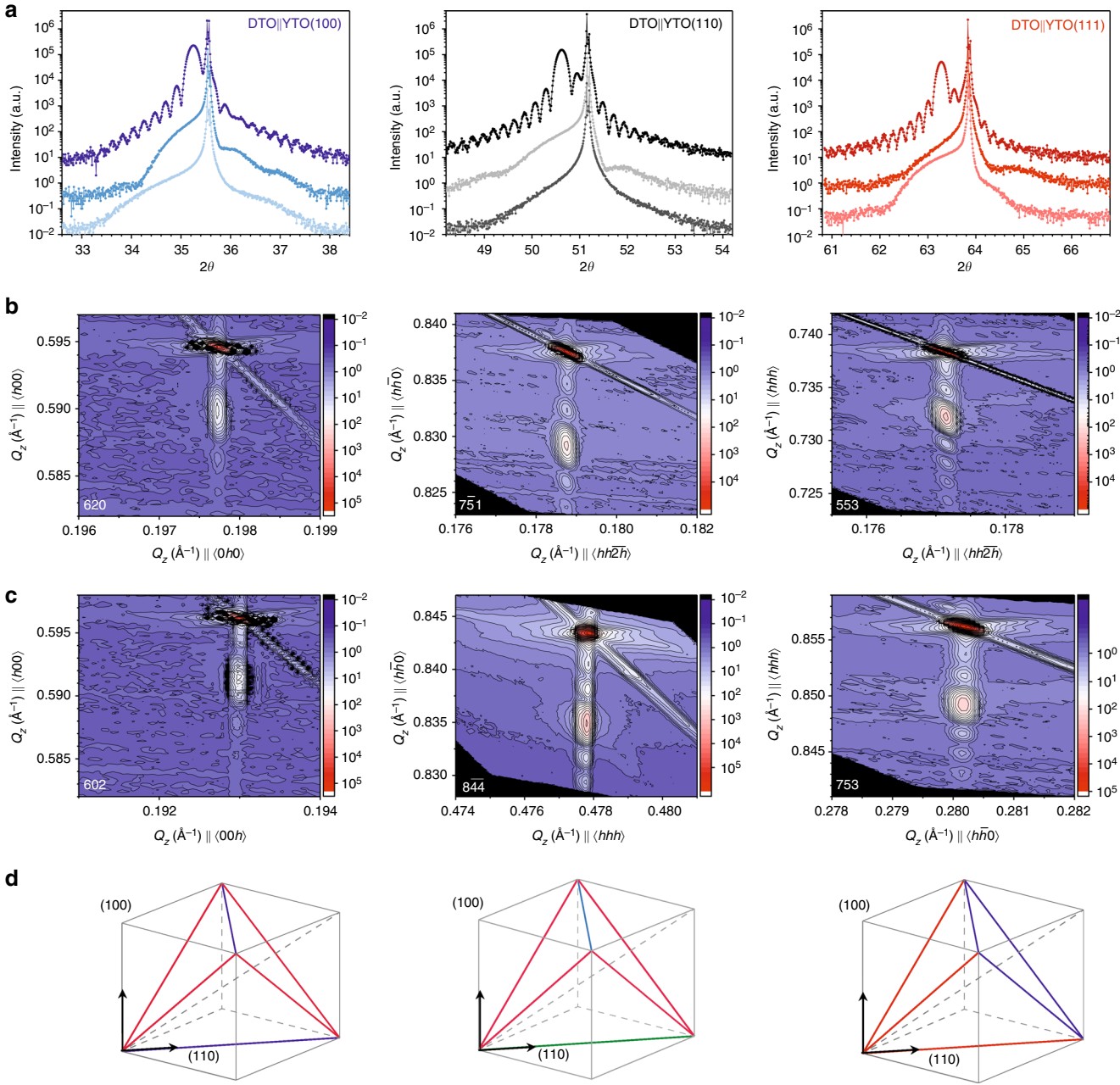

**Fig. 2** Epitaxy and compressive strain. Colour code: DTO||YTO(hkl) with (hkl) = (100) (blue), (hkl) = (110) (black), (hkl) = (111) (red). Each column reports data related to the same (hkl) as labelled in the first row. **a** High-resolution $2\vartheta - \omega$ scans showing, from left to right, the (400), (440) and (444) reflections for different film thicknesses. **b, c** Room temperature high-resolution X-ray powder diffraction (XRD) reciprocal space maps (RMS) for the thickest sample of each series. Here, the two orthogonal $\mathbf{Q_x}$ are the components of the reciprocal space vector aligned with the substrate in-plane edges. These RMS confirm the film is high quality, fully epitaxial, and fully strained along the two in-plane orientations. **d** Schematics of the Dy sub-lattice are shown, where the tetrahedra edges depict the Dy–Dy distances. In the pyrochlore cubic structure these are all identical. In each strained DTO||YTO(hkl) film, a different distortion is realised. Here, different colours represent different Dy–Dy distances (values are reported in Table 2). DTO $Dy_2Ti_2O_7$, YTO $Y_2Ti_2O_7$

that certain small features described below (hysteresis) were not as clearly seen, possibly due to the difficulties in quantitatively separating the contribution of the active material (DTO) from that of the substrate (YTO) for such thin layers.

For each series of samples, the magnetic specific heat, $C_M$, divided by temperature is plotted vs. temperature in Fig. 3. A bulk single crystal DTO has also been measured and reported in Fig. 3, for comparison. The behaviour of all thin films is close to that of bulk spin ice at $T > 2$ K, but starts to deviate from it at lower temperatures, as previously reported[34]. Furthermore, as shown in Fig. 3d, curves for samples grown along different crystallographic

orientation are coincident for $T > 0.75$ K, whereas at lower temperature each set of samples behaves differently.

For 44DTO||YTO(110) and 60DTO||YTO(111) (Fig. 3b, c, respectively), a small hysteresis loop opens up at low enough temperatures, below $T = 0.75$ K. During cooling, a small but clear and reproducible discontinuity is observed at $T = 0.510(2)$ K (44DTO||YTO(110)) and $T = 0.520(2)$K (60DTO||YTO(111)). In both samples, the width of the hysteresis loop does not vary for repetitive cycles. In contrast, for 44DTO||YTO(100), corresponding to Fig. 3a, data measured during cooling and heating do not show any hysteresis or discontinuity. In the Supplementary

Figure 1 we show how the present data are fully consistent with that of ref. [23], even though the latter failed to resolve these transitions. The observed hysteresis and orientation-dependent specific heat below the peak in the specific heat suggests an ordering transition. This transition must be of a very unusual sort as the anomaly is extremely weak. The 44 ML (100) film does not show any obvious transition, but as discussed below, a transition is visible in a much thinner film of this orientation.

**Comparison with vertex models**. These results may be compared with expectations for vertex models. Theoretical results that are broadly relevant to this situation include exact solutions for two-dimensional (2D) vertex models like the Slater[2] and F-models[3,5], in which the degeneracy of the ice-like six-vertex manifold is reduced such that a pair of vertices lies lowest in energy, see inset of Fig. 4a. In addition, a recent study[40] of the undistorted dipolar spin ice model in slab geometry identified a tendency to the F-model, with the complication of a surface phase transition. Slater

and F-type models are of particular interest as they are defined on a topologically constrained configurational manifold and exhibit ordering transitions from a critical phase. In the spin ice context, this is a Coulomb phase with monopole excitations[22], and transitions out of this state caused by perturbations may be viewed as Higgs transitions[41].

In detail, the cubic spin ice structure is homogeneously compressed within the plane causing differences in bond length on each spin tetrahedron. A film grown along [100] of the parent cubic structure has two shorter in-plane bonds and four longer out-of-plane bonds. In terms of vertex weights, this produces either an F- or a Slater-type model, but the former is the more likely for the following reasons. The near-neighbour coupling in spin ice consists of a dominant ferromagnetic dipolar term and a weaker antiferromagnetic exchange term. The latter will vary with distance much more rapidly than the former; hence, it is reasonable to expect that compressed (in-plane) bonds will be weaker than relaxed (out-of-plane) ones. The result is a splitting of the six-fold degenerate ground state of a spin tetrahedron in the manner of the F-model, rather than the Slater model. Furthermore, the distortions expected for films grown along [110] and [111] (surprisingly) do not break the six-fold degeneracy of spin ice at the level of nearest-neighbour interactions. Hence, we might expect the rough applicability of the results of ref. [40] to this case, with this time an effective F-model modified by dipole interactions.

Given that we expect the 2D F-model to be the basic model of a spin ice thin film at low temperature, it is worth recalling its properties. The basic model admits only six vertices and the high temperatures state tends to the Pauling state. The transition to an ordered phase[5,7,42], although ostensibly an order–disorder transition, is formally classified as an unusual representative of

---

**Table 2 DTO∥YTO(hkl) structural distortion with (hkl), the out-of-plane growth orientation**

| Sample | $d_{(110)}$ (Å) | $d_{(1\bar{1}0)}$ (Å) | Other orientations (Å) |
|---|---|---|---|
| Bulk DTO | 3.5815 | – | – |
| (100) | 3.567 (blue) | 3.567 (blue) | 3.583 (red) |
| (110) | 3.599 (green) | 3.567 (blue) | 3.575 (red) |
| (111) | 3.567 (red) | 3.589 (blue) | see colour code |

Colour labels in brackets refer to that used in Fig. 2 bottom panels
DTO Dy$_2$Ti$_2$O$_7$, YTO Y$_2$Ti$_2$O$_7$

---

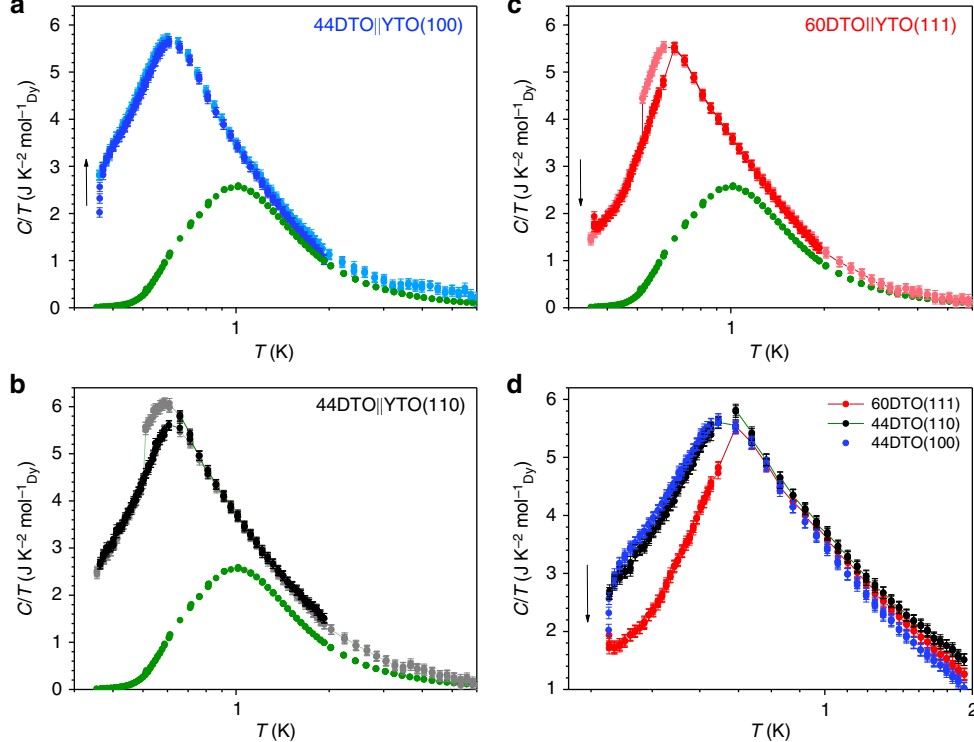

**Fig. 3** Specific heat. Bulk single crystal DTO (green) is also reported for comparison. Error bars represent estimated maximum systematic error. Note that for the bulk sample, error bars are within the symbol size. **a** 44DTO∥YTO(100), measured during cooling (light) and heating (dark). No hysteresis or discontinuity is visible in the temperature range explored. **b** 44DTO∥YTO(110), measured during cooling (light) and heating (dark). Here, a hysteresis loop opens up below $T = 0.75$ K and a sharp discontinuity is detected upon cooling at $T = 0.510(2)$ K. **c** 60DTO∥YTO(111), measured during cooling (light) and heating (dark). Here, a hysteresis loop opens up below $T = 0.75$ K and a sharp discontinuity is detected upon cooling at $T = 0.520(2)$ K. **d** Comparison between samples grown along different orientations, to emphasise how all the curves are superimposed above $T = 0.75$ K. DTO Dy$_2$Ti$_2$O$_7$, YTO Y$_2$Ti$_2$O$_7$

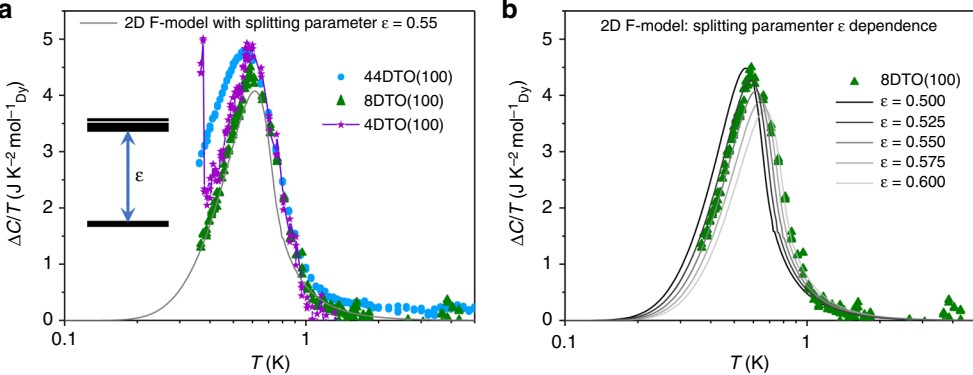

**Fig. 4** Two-dimensional (2D) F-model realisation. **a** Residual of the specific heat for the series of thin films xDTO||YTO(100) with x = 4, 8, 44 monolayers, after subtraction of the bulk DTO contribution. The full line (grey) is the exact solution for the 2D F-model with splitting parameter $\varepsilon = 0.55$ K. Inset shows energy splitting of the Pauling manifold, that is, from six degenerate states per tetrahedron to two ground states plus four excited states. **b** Splitting parameter dependence of the calculated 2D F-model residual specific heat (lines) plotted against 8DTO||YTO(100) as a guide to the eye. DTO $Dy_2Ti_2O_7$, YTO $Y_2Ti_2O_7$

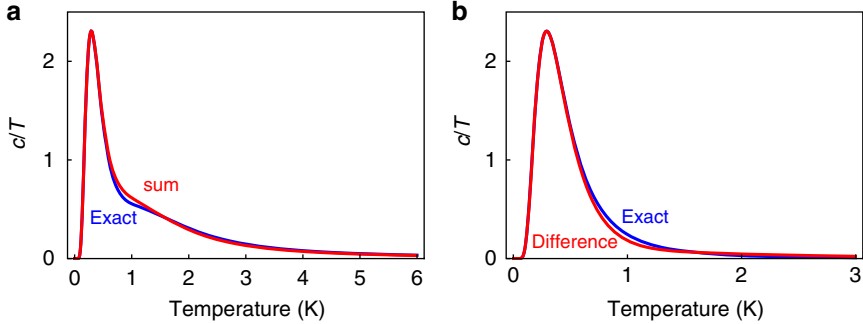

**Fig. 5** Subtraction Procedure for specific heat. **a** Comparison of the sum of the F-like and spin ice-like $(c/T)_{SI} + (c/T)_F$ (red) with the exact value $(c/T)_0$ (blue). **b** comparison of the difference $(c/T)_0 - (c/T)_{SI}$ (red) with $(c/T)_F$ for the F-like model (blue). Units of $c/T$ are J K$^{-2}$ mol$^{-1}_{Dy}$ (see Methods)

the Berezinskii–Kosterlitz–Thouless (BKT) class. The BKT transition, driven by the unbinding of topological defects (spin vortices or effective charges), more normally separates a high-temperature paramagnetic phase, with exponentially decaying correlations, from a low-temperature critical phase, with algebraically decaying correlations[43]. In the case of the F-model, the high-temperature six-vertex phase is critical, or algebraic, while the low temperature ordered phase has exponential correlations. At low temperature the entropy goes to zero, but the specific heat—which can be exactly calculated[5,7]—shows no anomaly. This 'non-scaling' behaviour has even been recommended as a surprising diagnostic of the transition[42]. The addition of extra vertices can change the nature of the transition, depending on the energy scales involved[44]. We note that 'non-scaling' and zero entropy are indeed properties of spin ice films[34].

In order to probe the relevance of the 2D F-model to the real films, we observe that in the limit of a weak perturbation to the Pauling manifold of the spin ice (16-vertex) model, the partition function of the system may be approximated by unperturbed (dipolar) spin ice at high temperature and by the F-model at low temperature. In this limit the specific heats of the F-model and spin ice model become additive (see Methods and Fig. 5). This approximation is not properly controlled for the actual films, where the perturbation is relatively strong. However, it would be expected to be accurate at low temperature where spin ice excited states are not yet thermally populated and cannot be too inaccurate at higher temperature as it correctly accounts for the total entropy increment (the integral of $C_M/T$—see Methods). Hence, we may assess the relevance of the 2D F-model by subtracting the

contribution of the spin ice excited states (the peak measured for bulk DTO) to the magnetic specific heat (divided by temperature) of the spin ice films and plotting the residual, as in Fig. 4.

Figure 4a compares the experimental result for films of varying thickness with the exact result for the 2D F-model with splitting parameter $\varepsilon = 0.55$ K (see Methods for the formula). It is seen that the 4 and 8 ML films closely approximate the 2D F-model prediction, especially in the low-temperature side of the peak in the specific heat, while the thicker (44 ML) film is slightly less well described (we believe that the slight vertical offset between the theory and experiment in the 44 ML data set may be attributed to small amounts of impurity in the substrate which give a finite contribution to the entropy). Figure 4b shows the effect of varying the splitting parameter ε: all the data are captured with the parameter in the narrow temperature range between 0.5 and 0.6 K; hence, we may quote $\varepsilon = 0.55 \pm 0.05$ K as an estimate of the splitting of the Pauling manifold. The small variation in estimated splitting could reflect our approximations or it could have a physical cause, such as a variation of strain, but we cannot distinguish these possibilities at present.

Both panels of Fig. 4 confirm our expectation that the 2D F-model describes the low temperature properties of the spin ice thin films to a close approximation. They suggest that a single-monolayer film (of tetrahedra) would indeed realise the 2D F-model. Our results show that such a monolayer film is available to experiment, although to measure it will be a challenge as thermodynamic probes will need to be extremely sensitive. More generally, Fig. 4 explains the puzzle implied by ref. [34]: how the films can release the spin ice entropy without an anomaly in the

specific heat. It now seems clear that the mechanism is related to that of the 2D F-model, which orders via an unusual BKT transition (in Fig. 4 this is indicated by a slight kink in the theoretical curve near 0.8 K, which is an artefact of truncating a summation: see Methods).

In the thinnest (4 ML) (100) film, a very striking jump in the specific heat is observed near to the lowest temperature that we can access (a similar jump is observed for the (111) film—see Supplementary Figure 2). We believe that this is a real effect, although given its unusual nature, we interpret it only with caution. Given that this feature is not observed clearly in the thicker films, and given that it has only a very small effect on the observed entropy (see Supplementary Figure 3), we might speculate that it signifies a surface phase transition; indeed, the very weak anomalies observed in some of the thicker films may also be in this class. The simple F-model envisaged by the stacking of identical spin tetrahedra puts a ferromagnetic moment perpendicular to the film direction that is likely to be removed by the re-ordering of surface charge to minimise magnetostatic energy. A tangible example of this interesting physics has been given in ref. [40], but we cannot compare this directly with the experiment as the model analysed in ref. [40] has a different effective symmetry (quasi-cubic) to the epitaxial spin ice films discussed here (orthorhombic, A1). Nevertheless, our results are not inconsistent with the general implications of ref. [40].

In conclusion, thin film epitaxy allows the possibility of realising theoretical models of spin ice in confined dimensionality[40,45], at interfaces[46,47] and in reduced symmetry. One route to realising the exotic physics of low symmetry vertex models is to apply very high pressure to bulk spin ice[48], but in practical terms, the very large homogenous distortions offered by epitaxial thin films offer a much more promising route. The films have the experimental advantage of being mechanically stable, which allows for diverse experimental probes, and given that their growth can be monitored layer by layer, they are less susceptible to uncontrolled defects and disorder than are bulk pyrochlore materials[33]. Our results show that it is possible to prepare single crystal spin ice films right down to the 2D limit of an ML of tetrahedra. The development of experimental methods to probe this limit would be of great interest. Similarly, we have presented evidence to suggests the F-model is the minimal theoretical model of the spin ice films and should be the starting point for any theoretical description. It would be of great interest to study such a model in the slab geometry of the real films and with perturbations appropriate to experiment.

## Methods

**Single crystal growth for substrates**. Single crystals of YTO were grown using optical floating-zone technique[26]. First stoichiometric powder sample was prepared using high-purity $Y_2O_3$ and $TiO_2$ chemicals, sintered at 1200 °C for 48 h with intermediate grinding. A cylindrical rod of diameter 12 mm was placed in an optical floating-zone furnace and grown into a single crystal at a rate of 3.5 mm h$^{-1}$ in Ar/$O_2$ mixed gas flow atmosphere.

**Pulsed laser deposition**. Single crystals of $Y_2Ti_2O_7$ were cut and epi-polished on one side (SurfaceNet GmbH, http://www.surfacenet.de). Three sets of fully oriented YTO substrates were prepared: YTO110K111, YTO111K110 and YTO100K010 where the first crystallographic direction ($hkl$) is the out-of-plane orientation and the second ($Khkl$) identify one of the in-plane edges of the square substrate. Epi-taxial DTO thin films (between 65 and 5 nm in thickness) were grown on YTO substrates by pulsed laser epitaxy (KrF, $\lambda = 248$ nm) at 750 °C in 113 mTorr $O_2$. The laser fluence at the target was fixed at 1.97 J cm$^{-2}$; the laser repetition rate was initially set at 1 Hz for the first 200 shots and increase to 5 Hz for the remaining growth. Samples were subsequently post-annealed for 1 h at 750 °C in 400 Torr $O_2$ before cooling down to room temperature. RHEED was used to monitor the surface structure and to control the film thickness with atomic-layer precision.

**Structural characterisation**. Lattice parameters and film epitaxy were studied at room temperature by XRD using Cu $K_\alpha 1$ radiation in a Rigaku high-resolution

diffractometer. Film thickness was determined by X-ray reflectivity; the fits were performed with the Integrated Thin Film Analysis Software GlobalFit 1.3 (Rigaku Corporation). High-resolution reciprocal space maps were collected using the same machine.

**Specific heat**. The specific heat of each sample was measured from 50 to 0.4 K with a Quantum Design PPMS (Physical Properties Measurement System) with a $^3$He option. The data were corrected for the addenda (sample holder, attaching grease) heat capacity, which was measured in a separate run. Furthermore, each sample was measured during both cooling and warming to check if any hysteretic behaviour could be observed.

**2D F-model**. The specific heat per mole of Dy is given by:

$$C_M = \frac{RT}{2} \left\{ \frac{-\partial^2 G_{low}}{\partial T^2} \left[ 1 - \theta\left(T - \frac{\varepsilon}{\log(2)}\right) \right] + \frac{-\partial^2 G_{high}}{\partial T^2} \theta\left(T - \frac{\varepsilon}{\log(2)}\right) \right\}, \quad (1)$$

where $\theta$ is the Heaviside theta, $\varepsilon$ is the splitting parameter and $G$'s are Gibbs free energy contributions given by:

$$G_{low} = -RT \left[ \frac{\lambda}{2} + \sum_{m=1}^{\infty} \frac{\exp(-\lambda m)\tanh(\lambda m)}{m} \right], \quad (2)$$

$$G_{high} = \frac{-RT}{4\mu} \int_0^{\infty} \frac{1}{\cosh\left(\frac{\pi x}{2\mu}\right)} \frac{\cosh(x) - \cosh(2\mu)}{\cosh(x) - 1} dx, \quad (3)$$

where

$$\lambda = \operatorname{arccosh}\left(\frac{1}{2}\exp\left(\frac{2\varepsilon}{T}\right) - 1\right), \quad (4)$$

$$\mu = \arccos\left(\frac{1}{2}\exp\left(\frac{2\varepsilon}{T}\right) - 1\right). \quad (5)$$

These expressions were evaluated numerically to give the different curves in Fig. 4.

**Subtraction procedure for specific heat**. The system approximates an F-model (2 + 4 vertex) at low temperature and a spin ice (6 + 8 + 2-vertex) model at high temperature. We argue that the specific heat divided by temperature is approximately the sum of these contributions. The accuracy of this approximation is perhaps best demonstrated by means of an example. We choose a single tetrahedron model of spin ice with energy levels at 0 K (6 levels), 4 K (8 levels) and 16 K (2 levels). The 6-fold degenerate ground term is then split such that two states are at 1 K lower than the other four. The partition function is (with energies measured in kelvin):

$$Q_0 = 2\exp\left(\frac{1}{T}\right) + 4 + 8\exp\left(\frac{4}{T}\right) + 2\exp\left(-\frac{16}{T}\right). \quad (6)$$

We also define the partition function of the unperturbed spin ice-like model,

$$Q_{SI} = 6 + 8\exp\left(\frac{4}{T}\right) + 2\exp\left(-\frac{16}{T}\right) \quad (7)$$

and the partition function of an F-like model,

$$Q_F = 2\exp\left(\frac{1}{T}\right) + 4 \quad (8)$$

From these we compute the corresponding specific heats $(c/T)_0$, $(c/T)_{SI}$ and $(c/T)_F$. In Fig. 5 we show how the sum $(c/T)_{SI} + (c/T)_F$ compares to the exact $(c/T)_0$ and how the difference $(c/T)_0 - (c/T)_{SI}$ compares to the exact $(c/T)_F$. It can be seen that the differences are always small. This roughly reproduces the experimental procedure which can be seen to be trustworthy for the values of the energy splitting we observe.

## Data availability

The datasets generated and/or analysed during the current study are available in the main text, the Supplementary information and from the co-corresponding author L.B. on reasonable request.

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

## Acknowledgements

We would like to thank R. Thorogate for technical assistance. We also thank M.J.P. Gingras and H. Kurebayashi for a related collaboration. Pulsed laser deposition of the thin films was conducted by L.B. and with assistance from CMR at the Centre for Nanophase Materials Sciences, which is a DOE Office of Science User Facility (CNMS2015-251). L.B. was supported by The Leverhulme Trust through the Early Career Fellowship programme (ECF2014-284). D.P. acknowledges support from the EPSRC grant EP/K028960/1. L.B. and S.T.B. acknowledge additional support from Leverhulme Trust grant RPG-2016-391.

## Author contributions

L.B. and S.T.B. conceived the project. D.P. prepared the target for PLD deposition and grew the crystals used to obtain substrates. L.B. grew the films with assistance from C.M. R. L.B. performed all the experiments and analysed the data. S.T.B. conceived the theoretical framework and calculated the 2D F-model. L.B. and S.T.B. wrote the paper and incorporated suggestions from the co-authors.

## Additional information

**Competing interests:** The authors declare no competing interests.

