## [Peer Review File · Nature Communications]

Reviewers' comments:

Reviewer #1 (Remarks to the Author):

The manuscript describes detailed and well-performed experiments on thin films of the spin ice material Dy₂Ti₂O₇. So far, only a few studies have been reported on spin ice thin films and it is of interest to see this further development.

The most novel aspect of the present study concerns the observation of a hysteresis in the specific heat below 0.75K and a discontinuity at 0.510K. In itself, these are interesting observations that merit scientific publication.

The authors interpret these features in relation to two-dimensional vertex models (the Slater and F-models) and suggest unconventional physics (an 'unusual BKT transition') to underlie their observations. In my opinion the discussion of these aspects should be made a bit more accessible to the non-expert reader and the rather profound claim for new, unconventional physics could perhaps be made stronger than just based on these small anomalies in specific heat experiments and their theoretical modelling? On the other hand, I am wondering about the effects of surfaces and potential off-stoichiometries in these ultra thin films on their magnetic behaviour. Can the authors rule out such more mundane causes for their observations?

I also noted a few smaller typographical errors:

Page 4: '.. is as an example ..' should be '.. is an example ..'

Page 11: the] behind Hurbutise] should be removed

Figure 4 right: 'paramenter' should be 'parameter'

Supplementary Materials: '.. grown in to a single ..' should be '.. grown into a single crystal ..'

Supplementary Materials: 'Three states ... was prepared' should be 'Three states ... were prepared'

Reviewer #2 (Remarks to the Author):

The authors present an experimental study of epitaxially grown thin films of spin ice, where the lattice distortion induces phase transitions seemingly of the BKT/F-model type. This is a very interesting advance in the technical capability of thin film growth of spin ice and related oxide materials, as well as a new scientific discovery, following recent theoretical proposals. This will be of interest to the related community as well as to a broader range of scientists working on thin films as well as constrained models and critical systems. I am happy to recommend the paper for publication in Nature Communications, provided that the authors address the major comments reported below.

In order of appearance, I think that the authors ought to address the following comments before the paper can be published:

1) on p.5 the authors make the claim that the <100> growth direction is fastest, and say that this may be related to the floating zone growth direction being <110>. I do not understand the logic: shouldn't the floating zone growth direction then be the <100> direction?

2) on p.6 the authors discuss the distortion dependence on the growth direction. Is there a dependence on the film thickness as well? A statement to answer this question should be included, even if possibly obvious from earlier work.

3) on p.9 the authors claim that the specific heats of the F-model and spin ice become additive in the limit of weak perturbation. They should either refer to a result in the literature or include a demonstration of the additivity, at least in the Suppl.Mat. and refer to it in the main text.

4) on p.10, the authors discuss the striking jump in the specific heat near the lowest temperature in the thinnest film, which they speculatively relate to a surface phase transition. I think it is important for the reader to understand how this jump relates to loss of entropy at that temperature. For this reason, it would be very important, at least in the Suppl.Mat. to add the relevant integrated entropy curves.

5) fig.1: "In the case of DTO||YTO(100) only one oscillation is detected": the authors ought to make this point clearer. I understand what the caption and main text say about the different periodicity in the different growth directions, however, I do not see at all how I am supposed to see this behaviour in the (right panel of the?) figure. As it stands, the statement seems to be unsupported by the data.

6) fig.4: I do not understand what the inset in the left panel is meant to tell the reader. The authors should explain in greater detail.

7) The authors should add, at least in the Suppl.Mat. the curves for the entropy (from integrating C/T) to show the loss of entropy in all cases studied in the paper. It would be very useful if the reader could see these curves and gauge the authors' claims about their behaviour.

8) the right panel in Fig.4 begs for a weighted average of the F-model heat capacities over a distribution of values of epsilon. I strongly urge the authors to try it, and -- depending on how well it works -- include it either in the Suppl.Mat. or in the main text figure. They could use a gaussian distribution for epsilon centred at 0.55 and of width 0.05. If the result matches the data substantially better than each individual value, then it is perhaps suggestive of a distribution of strains in the film? (which I think would not be unreasonable to expect anyway)

In addition to the key issues mentioned above, I spotted some minor ones that the authors may want to address before resubmission:

1) typo p.4 "in as an example" (remove "as"?)

2) p.4 "roughly 4 time[s] bigger"

3) on p.10, the authors say that epsilon ranges between 0.5K and 0.6K, and therefore they take it to be 0.5 ± 0.05 . Shouldn't it be 0.55 ± 0.05 ?

4) there may be a typo in the conclusions: "in Lantagne-Hurtubise]"

5) typo at the end of the paper: "It would be to great interest"  "of" ?

6) fig.1: "Two repetitions rate"  "Two repetition rates"

7) fig.1: the last sentence "In these plots, ... to improve clarity" refers to the right panel, not the left panel.

8) fig.4: "The line (black)" -- the line is not black in the left panel.

9) in the Suppl.Mat.: "corrected for the addenda heat capacity" -- I am not sure I understand what the authors mean by "addenda"

10) on p.24 there is a typo in one of the equations (missing ")"), and the last equation should be followed by a period.

11) Fig.S1 and S2 should be accompanied by a small paragraph in the Suppl.Mat. that refers to them.

12) Fig.S1: "although old ones" maybe add "the"

13) Fig.S2: "eye). Thinnest" maybe add "The"

14) Fig.S2: ", x=5 also shows" missing comma after "5"

Reviewer #3 (Remarks to the Author):

The authors describe their realization of spin ice films of Dy₂Ti₂O₇ (DTO) with different crystallographic directions. Based on the confrontation of several experimental technics (Xray, electronic diffraction), they first prove that their samples have an atomic scale resolution, with a well determined number of layers.

Then, using specific heat measurements at low temperatures, they show that the zero temperature entropy of the bulk material is not present here, due to a lift of the energy of the six vertices that were degenerated in the bulk material, giving a model very near the 2D F-model. An unusal BKT transition (of infinite order) occurs for the thinnest films, evidenced on the measurements.

Some remarks below:

- In the legend of Fig. 1, are the words 'left' and 'right' at the right places ? Could the authors detail the meaning of the three curves of each subfigure and the conclusions we can get from them ?
- In the conclusion, the authors claim that it is possible to engineer single layer of spin ice, but that some experimental limitations exist. What are they exactly ?

In conclusion, very interesting 2D materials have been created here, that will certainly lead to further developments in view of monolayers. But even as they are at present (4-5 monolayers), they are the starting point to interesting theoretical studies. I recommend publication of this article in Nature Communications.

Reviewers' comments:

Reviewer #1 (Remarks to the Author):

The manuscript describes detailed and well-performed experiments on thin films of the spin ice material Dy₂Ti₂O₇. So far, only a few studies have been reported on spin ice thin films and it is of interest to see this further development.

The most novel aspect of the present study concerns the observation of a hysteresis in the specific heat below 0.75K and a discontinuity at 0.510K. In itself, these are interesting observations that merit scientific publication.

We thank the reviewer for recognising the interest in our thin films work, for the careful reading of the manuscript and helpful comments.

The authors interpret these features in relation to two-dimensional vertex models (the Slater and F-models) and suggest unconventional physics (an 'unusual BKT transition') to underlie their observations. In my opinion the discussion of these aspects should be made a bit more accessible to the non-expert reader

We agree and have added new text to clarify this.

First, in the introduction we clarify the connection of spin ice with vertex models:

“In fact, the physics of bulk spin ice is even richer than that of a conventional sixteen-vertex model: the long-range part of the dipole-dipole interaction, which is significant in spin ice, is largely self-screened in the six-vertex spin ice state (22,23), but manifests as a magnetic Coulomb interaction between excited vertices to give the fractionalized ‘magnetic monopoles’ of spin ice (20,24). The discovery (20,22-24) that the addition of the long-ranged and conditionally convergent dipole-dipole interaction to a conventional vertex model resulted in such sharply-defined, exotic and realizable physics was a surprise that has stimulated considerable interest and activity in the field (21,25-32).”

Second in the text we discuss the unusual BKT transition:

“The transition to an ordered phase (6,41), although ostensibly an order-disorder transition, is formally classified as an unusual representative of the Berezinskii-Kosterlitz-Thouless (BKT) class. The BKT transition, driven by the unbinding of topological defects (spin vortices or effective charges), more normally separates a high-temperature paramagnetic phase, with exponentially decaying correlations, from a low-temperature critical phase, with algebraically decaying correlations (42). In the case of the F-model, the high-temperature six-vertex phase is critical, or algebraic, while the low temperature ordered phase has exponential correlations.”

and the rather profound claim for new, unconventional physics could perhaps be made stronger than just based on these small anomalies in specific heat experiments and their theoretical modelling?

We are able to do this because we are in a sense perturbing away from bulk spin ice, which is very well understood, so our model is reliable, and then the specific heat is THE signature of the unusual transition, as reported in the literature (paper by M. Weigel & W. Janke).

On the other hand, I am wondering about the effects of surfaces and potential off-stoichiometries in these ultra thin films on their magnetic behaviour. Can the authors rule out such more mundane causes for their observations?

Certainly, we can rule out such causes for the *main* shape of the specific heat peak, which signals approach to the unusual BKT transition: detailed characterisation of the films in our previous papers shows that the thin film samples are as at least as “clean” as the bulk. Regarding the small “jumps” in the specific heat, of course we cannot be quite so confident as they are small effects. Accordingly, we only devote little space to these features and are careful to make it clear that our interpretation is speculative. Therefore, if they do have a complex origin, that would not change our conclusions too much. We have slightly altered the text to make this clearer:

“ We believe that this is a real effect, although given its unusual nature, we interpret it only with caution. As this feature is not observed clearly in the thicker films, and as it has only a very small effect on the observed entropy (see Figure S4), we might speculate that it signifies a surface phase transition; “

I also noted a few smaller typographical errors:

Page 4: '.. is as an example ..' should be '.. is an example ..'

Page 11: the] behind Hurbutise] should be removed

Figure 4 right: 'paramenter' should be 'parameter'

Supplementary Materials: '.. grown in to a single ..' should be '.. grown into a single crystal ..'

Supplementary Materials: 'Three states ... was prepared' should be 'Three states ... were prepared'

All corrected – thank you!

Reviewer #2 (Remarks to the Author):

The authors present an experimental study of epitaxially grown thin films of spin ice, where the lattice distortion induces phase transitions seemingly of the BKT/F-model type. This is a very interesting advance in the technical capability of thin film growth of spin ice and related oxide materials, as well as a new scientific discovery, following recent theoretical proposals. This will be of interest to the related community as well as to a broader range of scientists working on thin films as well as constrained models and critical systems. I am happy to recommend the paper for publication in Nature Communications, provided that the authors address the major comments reported below.

We thank the Reviewer for the careful reading and critical comments.

In order of appearance, I think that the authors ought to address the following comments before the paper can be published:

1) on p.5 the authors make the claim that the $\langle 100 \rangle$ growth direction is fastest, and say that this may be related to the floating zone growth direction being $\langle 110 \rangle$. I do not understand the logic: shouldn't the floating zone growth direction then be the $\langle 100 \rangle$ direction?

This is a reference to the well-established fact that crystals, paradoxically, tend to be bounded by their *slowest* growing faces, at least when they have been grown in the absence of significant external forces. Also, we are simply trying to say that the observation is consistent with other known facts. We have tried to make this clearer by modifying the passage in question:

“This tendency to grow in layers of unit cells rather than tetrahedra suggests that $\langle 100 \rangle$ faces grow relatively quickly, as implied by the fact (see Ref. (12), Figure 2) that flux grown crystals show $\langle 111 \rangle$ faces (i.e. when grown in the absence of significant external forces crystals tend to eliminate their fastest growing faces). It also seems consistent with the observation that crystals grown by the floating zone technique (which encourages a particular growth direction) tend to show $\langle 110 \rangle$ preferential growth axes (36,37).”

2) on p.6 the authors discuss the distortion dependence on the growth direction. Is there a dependence on the film thickness as well? A statement to answer this question should be included, even if possibly obvious from earlier work.

To make this clear we have added the statement “All our studies have indicated that the distortion does not depend on film thickness.”

3) on p.9 the authors claim that the specific heats of the F-model and spin ice become additive in the limit of weak perturbation. They should either refer to a result in the literature or include a demonstration of the additivity, at least in the Suppl.Mat. and refer to it in the main text.

We agree with hindsight that this sounds mysterious. Also, our wording was careless. We have revised the main text as follows:

“...the partition function of the system may be approximated by unperturbed (dipolar) spin ice at high temperature and by the F-model at low temperature. In this limit the specific heats of the F-model and spin ice model become additive (see Supplementary Note 1 and Figure S3). This approximation is not properly controlled for the actual films, where the perturbation is relatively strong. However, it would be expected to be accurate at low temperature where spin ice excited states are not yet thermally populated and cannot be too inaccurate at higher temperature as it correctly accounts for the total entropy increment (the integral of C_M/T)”

Supplementary Note 1 and Figure S3 had been added accordingly.

4) on p.10, the authors discuss the striking jump in the specific heat near the lowest temperature in the thinnest film, which they speculatively relate to a surface phase transition. I think it is important for the reader to understand how this jump relates to loss of entropy at that temperature. For this reason, it would be very important, at least in the Suppl.Mat. to add the relevant integrated entropy curves.

In response to another referee, we have added a note of caution about interpreting this transition. However, we agree with the comment above and have added entropy curves in the Figure S4. However, there is little entropy loss at the transition, which also suggests it comes from only part of the sample (possibly the surface).

The revised passage reads:

“We believe that this is a real effect, although given its unusual nature, we interpret it only with caution. Given that this feature is not observed clearly in the thicker films, and given that it has only a very small effect on the observed entropy (see Figure S4), we might speculate that it signifies a surface phase transition;”

5) fig.1: "In the case of DTO | YTO(100) only one oscillation is detected": the authors ought to make this point clearer. I understand what the caption and main text say about the different periodicity in the different growth directions, however, I do not see at all how I am supposed to see this behaviour in the (right panel of the?) figure. As it stands, the statement seems to be unsupported by the data.

We believe the reviewer has misinterpreted the figure – RHEED patterns are displayed in the left panel only; the right panel shows X-ray reflectometry data.

DTO | YTO(100) RHEED pattern is shown on the top left graph (blue). Compared to the other two growth directions shown in the two graphs below, we can only detect one oscillation before the signal disappears into the noise of the measurement. The other two RHEED

patterns (black and red) show clear oscillations that persist with time. As such, our claim is supported by the data.

6) fig.4: I do not understand what the inset in the left panel is meant to tell the reader. The authors should explain in greater detail.

Thank you for pointing this out. The inset figure was not very clear and so we have improved it accordingly. We have added to the caption:

“Inset shows energy splitting of the Pauling manifold, i.e. from six degenerate states per tetrahedron to two ground states plus four excited states.”

7) The authors should add, at least in the Suppl.Mat. the curves for the entropy (from integrating C/T) to show the loss of entropy in all cases studied in the paper. It would be very useful if the reader could see these curves and gauge the authors' claims about their behaviour.

We have now added this in Figure S4 and reference it in the main text.

8) the right panel in Fig.4 begs for a weighted average of the F-model heat capacities over a distribution of values of epsilon. I strongly urge the authors to try it, and -- depending on how well it works -- include it either in the Suppl.Mat. or in the main text figure. They could use a gaussian distribution for epsilon centred at 0.55 and of width 0.05. If the result matches the data substantially better than each individual value, then it is perhaps suggestive of a distribution of strains in the film? (which I think would not be unreasonable to expect anyway)

We thank the Reviewer for this suggestion. However, it is clear that the very small level of variation (0.05) is most likely of the same order as that arising from our two approximations (i.e. the “subtraction” now discussed in the SI and approximation of the slab model by the 2D model.) For that reason, we don't think we can meaningfully adopt the suggestion without “over-analysing” the data. However, we have added a statement to cover this:

“The small variation in estimated splitting could reflect our approximations or it could have a physical cause, such as a variation of strain, but we cannot distinguish these possibilities at present”.

In addition to the key issues mentioned above, I spotted some minor ones that the authors may want to address before resubmission:

Thank you -- these have been corrected. Also see comments below:

1) typo p.4 "in as an example" (remove "as"?)

2) p.4 "roughly 4 time[s] bigger"

- 3) on p.10, the authors say that epsilon ranges between 0.5K and 0.6K, and therefore they take it to be 0.5 ± 0.05 . Shouldn't it be 0.55 ± 0.05 ?
- 4) there may be a typo in the conclusions: "in Lantagne-Hurtubise]" .
- 5) typo at the end of the paper: "It would be to great interest"  "of" ?
- 6) fig.1: "Two repetitions rate"  "Two repetition rates"
- 7) fig.1: the last sentence "In these plots, ... to improve clarity" refers to the right panel, not the left panel.
- 8) fig.4: "The line (black)" -- the line is not black in the left panel.
- 9) in the Suppl.Mat.: "corrected for the addenda heat capacity" -- I am not sure I understand what the authors mean by "addenda"

This has been clarified in the text as follows:

“ The data were corrected for the addenda (sample holder, attaching grease) heat capacity,”

- 10) on p.24 there is a typo in one of the equations (missing ")"), and the last equation should be followed by a period.
 - 11) Fig.S1 and S2 should be accompanied by a small paragraph in the Suppl.Mat. that refers to them.
- We thank the reviewer for this comment, but in this occasion we tend to disagree. Both figures are properly referred in the main text (as per editorial policy) – adding a paragraph would simply duplicate the information.
- 12) Fig.S1: "although old ones" maybe add "the"
 - 13) Fig.S2: "eye). Thinnest" maybe add "The"
 - 14) Fig.S2: ", x=5 also shows" missing comma after "5"

All corrected – thank you!

Reviewer #3 (Remarks to the Author):

The authors describe their realization of spin ice films of Dy₂Ti₂O₇ (DTO) with different crystallographic directions. Based on the confrontation of several experimental technics (Xray, electronic diffraction), they first prove that their samples have an atomic scale resolution, with a well determined number of layers.

Then, using specific heat measurements at low temperatures, they show that the zero temperature entropy of the bulk material is not present here, due to a lift of the energy of the six vertices that were degenerated in the bulk material, giving a model very near the 2D F-model. An unusual BKT transition (of infinite order) occurs for the thinnest films, evidenced on the measurements.

We thank the Reviewer for the careful reading and comments.

Some remarks below:

- In the legend of Fig. 1, are the words 'left' and 'right' at the right places ?

Thank you – corrected.

Could the authors detail the meaning of the three curves of each subfigure and the conclusions we can get from them ?

We have modified the caption of the figure, to improve clarity. It now reads:

“Right,: Reflectivity measurements for all the xDTO | YTO(hkl) samples and related fits (values are reported in Table 1). In these reflectivity plots, each scan (samples with same hkl orientation, but different number of monolayers x) has been displaced vertically to improve clarity.”

- In the conclusion, the authors claim that it is possible to engineer single layer of spin ice, but that some experimental limitations exist. What are they exactly?

We have added the text:

“although to measure it will be a challenge as thermodynamic probes will need to be extremely sensitive.”

In conclusion, very interesting 2D materials have been created here, that will certainly lead to further developments in view of monolayers. But even as they are at present (4-5 monolayers), they are the starting point to interesting theoretical studies. I recommend publication of this article in Nature Communications.

We thank the Reviewer for the positive assessment.

REVIEWERS' COMMENTS:

Reviewer #1 (Remarks to the Author):

The authors have improved the manuscript according to my, and the other referee's, comments.

I am satisfied with the final result, which nicely presents the power of using epitaxial strain in realising and studying novel physical systems (in this case by turning a 3D spin ice compound effectively into a 2D-like system).

I recommend the manuscript to be published as it is.

Hans Hilgenkamp
University of Twente
The Netherlands

Reviewer #2 (Remarks to the Author):

The authors have addressed all my comments and criticisms and I am therefore happy to confirm my recommendation for the manuscript to be published in your journal